# Cross-Talk between Wnt Signaling and Src Tyrosine Kinase

**DOI:** 10.3390/biomedicines10051112

**Published:** 2022-05-11

**Authors:** Jung Ki Min, Hwee-Seon Park, Yoon-Beom Lee, Jae-Gyu Kim, Jong-Il Kim, Jae-Bong Park

**Affiliations:** 1Department of Biochemistry, Hallym University College of Medicine, Chuncheon 25242, Korea; jkmin0306@hallym.ac.kr (J.K.M.); barca9118@hallym.ac.kr (Y.-B.L.); mip11@hallym.ac.kr (J.-G.K.); 2Institute of Cell Differentiation and Aging, Hallym University College of Medicine, Chuncheon 24252, Korea; 3Department of Biomedical Sciences, Seoul National University College of Medicine, Seoul 03080, Korea; hwee38@snu.ac.kr (H.-S.P.); jongil@snu.ac.kr (J.-I.K.); 4Genomic Medicine Institute, Medical Research Center, Seoul National University College of Medicine, Seoul 03080, Korea

**Keywords:** Wnt, Src, GSK-3β, β-catenin, Rho GTPases

## Abstract

Src, a non-receptor tyrosine kinase, was first discovered as a prototype oncogene and has been shown to critical for cancer progression for a variety of tissues. Src activity is regulated by a number of post-translational modifications in response to various stimuli. Phosphorylations of Src Tyr419 (human; 416 in chicken) and Src Tyr530 (human; 527 in chicken) have been known to be critical for activation and inactivation of Src, respectively. Wnt signaling regulates a variety of cellular functions including for development and cell proliferation, and has a role in certain diseases such as cancer. Wnt signaling is carried out through two pathways: β-catenin-dependent canonical and β-catenin-independent non-canonical pathways as Wnt ligands bind to their receptors, Frizzled, LRP5/6, and ROR1/2. In addition, many signaling components including Axin, APC, Damm, Dishevelled, JNK kinase and Rho GTPases contribute to these canonical and non-canonical Wnt pathways. However, the communication between Wnt signaling and Src tyrosine kinase has not been well reviewed as Src regulates Wnt signaling through LRP6 tyrosine phosphorylation. GSK-3β phosphorylated by Wnt also regulates Src activity. As Wnt signaling and Src mutually regulate each other, it is noted that aberrant regulation of these components give rise to various diseases including typically cancer, and as such, merit a closer look.

## 1. Introduction

The single viral gene of avian sarcoma virus, v-src, was shown to have protein tyrosine kinase activity and induce neoplastic transformation of the cell. History of Src oncogene discovery and its study has been well reviewed by Simatou et al. [1]. By analogy, in normal chicken cells, c-src gene was found be related to the viral src gene [2]. V-src lacks the C-terminal peptide of chicken Src containing 10 amino acids including Tyr527 that is critical for Src inactivation. By missing the Tyr527-containing peptide, v-src is constitutively active, and its activity results in tumorgenicity in host cells [3,4]. After the finding of mammalian versions of Src, several homologous Src proteins have been described that include the Src family of kinases (SFKs) consisting of 10 proteins: Src, Frk, Lck, Lyn, Blk, Hck, Fyn, Yrk, Fgr, and Yes. As a family, the SFK family of tyrosine kinases are involved in a variety of key cellular processes. 

For the Wnt signaling pathway, since the discovery of the *wingless* gene in *Drosophila melanogaster* about 40 year ago, the pathways and the functions for Wnt have been extensively studied with Wnt signaling being critically involved in regulation of development, cell proliferation, migration, transcription and cancer formation. There are multiple target genes regulated by Wnt signaling, and to identify these target genes, three strategies can be generally applied: computational search for TCFs/LEFs (T cell factors/lymphoid enhancer factors), transcriptome analyses of Wnt-regulated genes, and chromatin immunoprecipitation sequencing analyses of TCF and β-catenin genomic occupancy. Wnt target genes have been reviewed in the following site: http://web.stanford.edu/group/nusselab/cgi-bin/wnt/ (accessed on 28 February 2022). 

There are two types of Wnt signaling pathways: the β-catenin dependent, canonical pathway and the β-catenin independent, non-canonical one. In the canonical pathway, for Wnt signaling through the Frizzled receptor (Fzd) and the low density lipoprotein (LDL) receptor-related protein 5/6 (LRP5/6), the activity of nuclear β-catenin is largely mediated by TCFs/LEFs. However, a subset of β-catenin transcriptional targets do not depend on TCF/LEF factors for their regulation, as seen when TCF/LEF factors were genetically knocked out in HEK 293T cells [5]. In β-catenin-independent noncanonical Wnt signaling pathway through Fzd receptors and receptor tyrosine kinase-like orphan receptor (ROR)1/ROR2/receptor tyrosine kinase (RYK) co-receptors, Wnt activates Rho GTPases including Rac1, RhoA, and c-Jun N-terminal kinase (JNK) that control rearrangements in the cytoskeleton and gene expression; these lead to the planar cell polarity (PCP) in morphogenetic processes in vertebrates. 

There is also the β-catenin-independent noncanonical pathway that is the WNT-Ca^2+^-dependent signaling cascade [6,7]. In particular, R-spondins are secreted proteins, implicated in the activation of Wnt signaling pathway. R-spondins bind to the orphan G-protein-coupled receptors LGR4 and LGR5, which promote R-spondin-mediated Wnt/β-catenin and Wnt/PCP signaling [8]. All R-spondins require Wnt ligands and LRP6 for activity and amplify signaling of Wnt3a, Wnt1 and Wnt7a [9]. Functionally, Wnt proteins cannot induce self-renewal of Lgr5+ intestinal stem cells, but instead confer a basal competency by maintaining R-spondin receptor expression that enables R-spondin ligands to actively drive and specify the extent of stem cell expansion [10]. There are also extracellular Wnt inhibitors including secreted Frizzled-related proteins (SFRPs), Wnt inhibitory factor-1 (WIF-1), Sclerostin and Dickkopf-related protein 1 (DKK1) that regulate Wnt signaling [11]. 

Aberrant Wnt signaling as well as increased Src activity is closely related to many diseases. In particular, the role of Wnt signaling in cancer progress has most prominently been described for colorectal cancer [12]. In the tumor microenvironment, WNT5A significantly reduces the number of cytotoxic T cells, decreases the presence of M1 macrophages and increases that of M2 macrophages in tumors, leading to immunosuppression in favor of the tumor [13]. Thereby, the adequate controls of aberrant Src activity and Wnt signaling pathway are crucially required for treatment of many diseases including typically cancers. 

## 2. Src Non-Receptor Tyrosine Kinase

Src non-receptor tyrosine kinase phosphorylates a variety of protein substrates that perform specific cellular functions. Activity of Src is regulated by a variety of stimuli and the Src protein is subjected to several types of post-translational modifications including lipidation, phosphorylation, acetylation, ubiquitylation, sumoylation and oxidation. In particular, p-Tyr416 Src has been known to be an active form while p-Tyr527 Src is an inactive form through autoinhibition by binding to Src SH2 own domain (Figure 1). Src also plays specific roles in specific compartments of the cell as Src is localized to the membrane, cytosol, mitochondria, and nucleus for specific functions [14]. Here, we discuss the regulation of Src activities concurrent with post-translational modification (Table 1).

### 2.1. Lipidation

Myristic acid is covalently linked to Gly2 instead of the Met initiator amino acid in Src. This myristoylation of Src is required for its membrane association, and cell transformation. Furthermore, the first 14 amino acids of Src contain a recognition sequence for its myristoylation [15,16]. Myristoylation is carried out by N-myristoyl transferase (NMT) by using myristoyl-CoA, with the consensus sequence for NMT protein substrate being Met-Gly-X-X-X-Ser/Thr. The initiating Met is cleaved by methionine amino peptidase and Gly2 becomes the N-terminal amino acid of the myristoylated Src [17].

Myristoylation stimulates Src activity by anchoring it to lipid membranes and the membrane binding also regulates Src ubiquitylation and degradation, as a nonmyristoylated Src has reduced activity, but it has enhanced stability. Src also has a hydrophobic pocket domain in the C-terminal region that binds to the myristoyl group. T456A mutation of Src leads to detachment from the lipid membrane, suggesting that Thr456 is localized to a binding pocket; this mutation also seems to promote to a nonmyristoylation state [18]. The N-terminal myristoylated SH4 domain of Src was recently reported to interact with its SH3 domain when Src is not anchored to a lipid membrane, and the residues in the domain termed the Unique lipid binding domain modulate this interaction [19]. For Src, in addition to its myristoyl group, six basic residues in the amino terminus (Myristoyl-GSSKSKPKDPSQRRR) are particularly essential for high affinity binding to the lipid bilayer via electrostatic interaction with acidic phospholipid, shown in vitro [20]. 

Other SFKs including Yes, Fyn, Lyn, Lck, Hck, Fgr, and Yrk harbor cysteine residue(s) in their N-terminal domain that can be covalently linked to palmitic acid by palmitoyl acyltransferase (PAT); this enzyme recognizes Met-Gly-Cys at the N-termini of these SFKs. However, Src does not contain this cysteine residue, and thus cannot be modified with palmitoylation [17].

### 2.2. Phosphorylation

#### 2.2.1. Phosphorylation of Tyr416 and Tyr527

Similar to most protein kinases, Src family kinases also require phosphorylation for regulation of their enzyme activity. Autophosphorylation of Src at Tyr416 (chicken; Tyr419 in human) ensures an active form [21]. In contrast, phosphorylation of Src at Tyr527 (chicken; Tyr530 in human) in the Src C-terminal domain inactivates the enzyme [22]. Tyr527 phosphorylation is carried out by C-terminal Src kinase (Csk) [23] and its homolog, Csk homologous kinase (Chk) [24,25]. Phosphorylation of the C-terminal Tyr527 of Src promotes binding of this phosphorylated residue to its own SH2 domain, resulting in an intramolecular auto-inhibitory Src structure [26,27]. Regarding the control of Src activity via Tyr527, receptor protein tyrosine phosphatase α (RPTPα) -/- mouse has impaired Src activity, manifested by a concomitant increase in phosphorylated Tyr527 forms of Src [28,29]. In addition, PDGF receptor, being a membrane tyrosine kinase, is able to phosphorylate Src at its Tyr213 residue near the binding pocket of SH2 domain, which then interferes with p-Tyr527 residue binding to the Src SH2 domain, leading to Src activation [30]. A detailed three-dimensional structure of Src with phosphorylations of Tyr416 and Tyr527 has also been described [31]. 

#### 2.2.2. Phosphorylation of Ser17, Ser37, Ser69, and Ser75 in the Unique Domain of Src

Src protein has several domains including SH1 (catalytic), SH2, SH3, SH4, and Unique domains. The Src Unique domain localized in the N-terminal region of the protein exhibits strong sequence divergence from the other SFK members. Recently, the Unique domain was reported to reveal a crucial role in regulation of Src activity. In particular, phosphorylation/dephosphorylation observed in the Unique domain of Src plays a critical regulatory function in Src kinase activity. There is also the unique lipid binding region (ULBR) that is a partially structured region within the Unique domain of Src [32]. 

Platelet-derived growth factor (PDGF) activation of the cells causes Src to translocate from the plasma membrane to the cytosol and a 4-fold activation of its kinase activity and phosphorylation of its N-terminal region. Ser17 of Src was postulated to be a candidate phosphorylation residue, which likely changes the hydrophobicity of Src [33]. Ser17 of Src was demonstrated to be phosphorylated by protein kinase A (PKA) in PC12 cells. Src S17A, its dephosphorylated mimic, inhibits Rap1-dependent ERK activation by NGF and cAMP. In addition, neurite outgrowth from PC12 cells by cAMP is also inhibited by Src S17A, suggesting that Ser17 phosphorylation of Src is required for neurite outgrowth by cAMP [34]. In addition to Ser17 phosphorylation by PKA, Thr37 and Ser75 (human; Thr34 and Ser72 in chicken, respectively) and Ser46 (chicken; not found in human) of Src are phosphorylated by Cdk1/Cdc2 during mitosis. Phosphorylations of Thr34, Thr46, and Ser72 residues by p34Cdc2 either sensitize chicken Src to a Tyr527 phosphatase or desensitize Src to a Tyr527 kinase, leading to Src activation [35]. 

On serine and threonine phosphorylated residues, phosphorylation of Thr37 (human; Thr34 in chicken) and Ser75 (human; Ser72 in chicken) by p25-Cdk5 attenuates lipid binding by the ULBR. Phosphorylation of Ser17, Ser37 and Ser75 of Src gives rise to an electric static repulsion against negative charged lipids within membrane, leading to disruption of Src-membrane interaction [36,37]. 

### 2.3. Acetylation

CREB binding protein (CBP) acetylates the N-terminal lysine residues Lys5, Lys7 and Lys9 of Src to promote dissociation from the cell membrane. In addition, CBP also acetylates the C-terminal Lys401, Lys423, and Lys427 of Src to activate intrinsic kinase activity for STAT3 recruitment and activation, resulting in N-terminal domain phosphorylation (Tyr14, Tyr45, and Tyr68) of STAT3. These phosphorylations of STAT3 lead to formation of STAT3 dimer, an active transcription factor, which translocates to nucleus and then regulates transcription of specific genes there. 

### 2.4. Ubiquitylation

Loss of Csk reduces Src and the related SFK member, Fyn, protein levels. This is due to Csk stabilizing Src levels, along with inhibition of proteasome activity also increasing Src protein levels in Csk-deficient cells, pointing to protein degradation by a proteasome as the possible mechanism in this regulation. Tyr419-phosphorylated Src, an active form, can undergo Cullin-5-dependent ubiquitylation and activation of Src increases the extent of its polyubiquitylation [38]. Moreover, phosphorylation of Ser75 by Cdk5 promotes the ubiquitin-dependent degradation of Src [39]. Interestingly, ubiquitylated Src at Lys429 is crucial for Src secretion via extracellular vesicles. Mutation of Src at Lys429 also activates the tyrosine kinase FAK to potentiate Src-induced invasive phenotypes [40]. Ubiquitin ligase subunit, FBXL7, mediates the ubiquitylation and proteasomal degradation of active Src after phosphorylation of Src Ser104 residue. It should be noted that the promoter of FBXL7 is hypermethylated in advanced prostate and pancreatic cancers along with decreased mRNA and protein levels of FBXL7 [41]. 

### 2.5. SUMOylation

Src can be SUMOylated at Lys318 in response to hydrogen peroxide. In contrast, hypoxia attenuates Src SUMOylation, along with an increase in Src Y419 phosphorylation. Ectopic expression of SUMO-defective mutant, Src K318R, promotes tumorigenesis. In addition, Src SUMOylation decreases Y925 phosphorylation of tyrosine kinase FAK residue, leading to reduced cell migration. Consequently, SUMOylation of Src at Lys318 negatively modulates its oncogenic function by at least partially inhibiting Src-FAK complex activity [42]. 

### 2.6. Oxidation

Intramolecular disulfide bridge formation between Cys245 and Cys487 upon exposure to ROS leads to Src activation [43]. In contrast, intermolecular disulfide bridges between the Cys277 residues of two different Src proteins result in inactive Src dimers [44]. Src family kinases localized to focal adhesion and the plasma membrane are rapidly and permanently inactivated by hydrogen peroxide. Surprisingly, the levels of cytoplasmic Src family kinases also gradually rise by hydrogen peroxide, in human aortic endothelial cells (HAECs) and human umbilical vein endothelial cells (HUVECs) [45]. All the post-translational modifications of Src are summarized in Figure 2. 

**Table 1 biomedicines-10-01112-t001:** Post-translational modifications of Src.

Posttranslational Modification	Description	References
Lipidation	Myristic acid is covalently linked to Gly2.	[15]
	The first 14 amino acids of Src contain a recognition sequence for myristoylation of Src	[16]
	The initiating Met is cleaved by methionine amino peptidase and Gly2 become to N-terminal amino acid.	[17]
	T456A mutation of Src undergoes detached from membrane, suggesting Thr456 is localized in binding pocket and regulates myristoyl switch.	[18]
	N-terminal myristoylated SH4 domain interact with SH3 domain when Src in not anchored to a lipid membrane.	[19]
	Myristoyl group is particularly essential for high affinity binding to the lipid bilayer via electrostatic interaction with acidic phospholipid in vitro	[20]
	SFKs in N-terminal domain and cysteine(s) can be covalently linked to palmitic acid by palmitoyl acyl transferase (PAT).	[17]
Phosphorylation	Tyr416 ensures an active form.	[21]
	Tyr527 in C-terminal domain reveals inactivation.	[22]
	The Tyr527 phosphorylation is carried out by C-terminal Src kinase.	[23]
	The Tyr527 phosphorylation is carried out by its homolog Csk homologous kinase (Chk).	[24,25]
	Tyr527 of Src promotes assembly SH2 domain of Src, resulting in intramolecular auto-inhibitory Src structure.	[26,27]
	PDGF phosphorylates Tyr213 which interferes with p-Tyr527 binding to SH2 domain in Src leading Src activation.	[30]
	Three-dimensional structure of Src with phosphorylations of Tyr416 and Tyr527.	[31]
	Phosphorylation/dephosphorylation observed in the Unique domain of Src plays a critical regulatory function in Src kinase activity.	[32]
	Ser17 of Src was postulated to be a candidate of phosphorylation residue, which likely changes hydrophobicity of Src.	[33]
	Ser17 of Src was demonstrated to be phosphorylated by PKA in PC12 cells. Src S17A (dephospho-mimic) inhibits Rap1-dependent ERK activation by NGF and cAMP.	[34]
	The phosphorylations of the Thr34, Thr46, and Ser72 residues by p34Cdc2 either sensitize chicken Src to a Tyr527 phosphatase or desensitize Src to a Tyr527 kinase, leading to Src activation.	[35]
	The phosphorylation of Ser37 and Ser75 by p25-Cdk5 attenuates lipid binding by the ULBR.	[36,37]
	Ser109 phosphorylation of Src undergoes its degradation.	[41]
	Wnt3A induces phosphorylation of Src at Ser43, Ser51 and Ser493 residues through p-Ser9 GSK-3β.	[46]
Acetylation	CREB binding protein (CBP) acetylates N-terminal lysine residues (K5, K7 and K9) of c-Src to promote dissociation from the cell membrane. In addition, CBP also acetylates the c-terminal K401, K423, and K427 of c-Src.	[47]
	Low aggressive osteosarcoma SaOS-2 cells show high level of Src in nucleus. High metastatic 143B osteosarcoma cells present low levels of nuclear Src.	[48]
	EGF induces SRC activation, which phosphorylates Tyr845 in EGFR, resulting in mitochondrial localization of Src and EGFR.	[49]
	In mitochondria, EGFR binds to cytochrome C oxidase subunit II (CoxII), and EGFR and Src phosphorylate CoxII, leading to decreases of complex IV activity and ATP levels.	[50]
	Src is sequestered to mitochondria with AKAP121.	[51]
	Overexpression of the downstream of kinase-4 (Dok-4) containing N-terminal mitochondrial targeting sequence increases mitochondrial Src localization through the complex formation with Src.	[52]
	Mitochondrial Src is high in breast cancer cells of triple negative subtype, and targets to phosphorylate mitochondrial single stranded DNA-binding protein (SSBP1), a regulator of mtDNA replication.	[53]
Ubiquitylation	p-Tyr419 Src, an active form can undergo Cullin-5-dependent ubiquitylation and activation of Src increases the extent of polyubiquitylation.	[38]
	Phosphorylation at Ser75 by Cdk5 promotes the ubiquitin-dependent degradation of Src.	[39]
	Mutation at Lys429 activates FAK to potentiate Src-induced invasive phenotypes.	[40]
	The promoter of FBXL7 is hypermethylated in advanced prostate and pancreatic cancers along with decreased mRNA and protein levels of FBXL7.	[41]
SUMOylation	SUMOylation of Src at K318 negatively modulates its oncogenic function by at least partially, inhibiting Src-FAK complex activity.	[42]
Oxidation	Intramolecular disulfide bridge between Cys245 and Cys487 upon exposure to ROS lead to Src activation.	[43]
	Intermolecular disulfide bridges between Cys277 residues of two different Src proteins result in inactive Src dimers.	[44]
	Cytoplasmic Src family kinases are activated gradually by hydrogen peroxide in human aortic endothelial cells (HAECs) and human umbilical vein endothelial cells (HUVECs).	[45]

## 3. Localization of Src in Cell Compartments

### 3.1. Translocation of Src to Nucleus

Notably, acetylated Src at Lys5, Lys7, and Lys9 translocates into the nucleus, where it associates with STAT3 for specific gene regulation and cancer cell proliferation. Furthermore, acetylated Src at Lys401, Lys423 and Lys427 increases intrinsic kinase activity for STAT3 recruitment and activation [47]. Low aggressive osteosarcoma SaOS-2 cells show high levels of Src in the nucleus along with Src having low myristoylation and the cells being low in expression of N-myristoyltransferase (NMT) enzyme. In contrast, the highly metastatic 143B osteosarcoma cells present low levels of nuclear Src, being high myristoylated, and having high levels of NMT [48]. 

### 3.2. Translocation of Src to Mitochondria

EGF induces Src activation, which then phosphorylates Tyr845 of EGFR, resulting in mitochondrial localization of Src and EGFR. Inhibition of Src completely prevents translocation of EGFR and Src to the mitochondria [49]. In mitochondria, EGFR binds to cytochrome C oxidase subunit II (CoxII), leading to phosphorylation of CoxII and decreased complex IV activity and ATP levels [50]. A-kinase anchor protein 121 (AKAP121), which anchors to protein kinase A (PKA), associates with protein tyrosine phosphatase D1 (PTPD1) and Src in addition to PKA. PTPD1 binds to and activates Src on AKAP121, which ensures PTPD1 being sequestered to mitochondria. This behavior may also be viewed as Src being sequestered to mitochondria with AKAP121 [51]. 

Overexpression of the downstream of kinase-4 (Dok-4) containing an N-terminal mitochondrial targeting sequence increases mitochondrial Src localization through complex formation with Src. In mitochondria, Src downregulates the level of mitochondrial complex I, leading to increased ROS production in the sequence of Dok-4/Src/complex I/ROS events. Dok-4 also induces TNF-α-mediated NF-κB activation, suggesting that Src in mitochondria is critical for NF-κB activation through ROS production [52]. 

Mitochondrial Src is high in breast cancer cells of triple negative subtype, and this Src phosphorylates mitochondrial single stranded DNA-binding protein (SSBP1), a regulator of mtDNA replication. Mitochondrial Src thus reduces mtDNA levels, mitochondrial membrane potential and cellular respiration; these result in a shorter cell cycle with reduced cell viability and ramped up invasive capability [53]. 

## 4. Wnt Signaling

### 4.1. Canonical Wnt Signaling

In the β-catenin dependent canonical pathway of Wnt signaling, glycogen synthase kinase 3 (GSK-3) plays a crucial role for β-catenin regulation [54]. Mammals express two GSK-3 isoforms, α at 51 kDa and β at 46 kDa in molecular weight. Both GSK-3 isoforms are likely to function redundantly in certain signaling pathways, but they play distinct roles in certain others. Regulation of GSK-3 activity in Wnt and insulin signaling pathways has been well established [54]. In general, there are two pools of β-catenin in cells. One pool of β-catenin associates with cadherin in adherence junctions and the other pool exists in cytosol and nucleus. In a resting and unstimulated state, there is the β-catenin destruction complex, comprising of Axin, adenomatous polyps coli (APC), casein kinase 1 (CK1), GSK-3 and β-catenin. In the canonical Wnt signaling pathway, in this complex, CK1 phosphorylates Axin, APC, and β-catenin at Ser45 for ‘priming’ and then GSK-3β subsequently phosphorylates β-catenin at Ser33, Ser37, and Thr41, thereby leading to β-catenin proteasomal degradation through β-TrCP (E3 ubiquitin ligase subunit). In this complex, the activity of GSK-3β coupled to Axin is not regulated by phosphorylation. This independence of regulation by GSK-3β phosphorylation may be attributed to structural barriers, which interfere with access of other kinases to phosphorylate GSK-3β [6]. For canonical Wnt (Wnt2, Wnt3, Wnt3a) signaling, Fzd receptors and LRP5/LRP6 co-receptor transduce Wnt signals to the β-catenin signaling cascade. CK1 with a membrane anchor in the form of a palmitoylation domain phosphorylates the LRP cytoplasmic domain, particularly the LRP tail, which then recruits and binds to Axin protein. As GSK-3β cannot phosphorylate β-catenin, it ensures an increase of cytoplasmic β- catenin levels with β-catenin subsequently translocating to the nucleus and inducing gene expression in complex with TCF/LEF transcription factors [7]. Furthermore, the phosphorylated PPP (1490) SPxS motif in the intracellular region of LRP5/6 recruits, binds to and directly inhibits GSK-3β activity in Wnt/β-catenin signaling, which is another mechanism for β-catenin accumulation in Wnt signaling [55]. The crystal structures of GSK-3 bound to its phosphorylated N-terminus and to two of the phosphorylated LRP6 motifs have been solved, and a conserved loop unique to GSK-3 was found to undergo a dramatic conformational change that clamps the bound pseudo-substrate peptides, presenting a mechanism of primed substrate recognition [56]. 

### 4.2. Non-Canonical Wnt Signaling

In the β-catenin-independent noncanonical Wnt signaling, Fzd receptors assemble with ROR1/ROR2/RYK co-receptors and transduce the Wnt signal to Rho GTPases including Rac1 and RhoA and JNK that control rearrangements in the cytoskeleton and gene expression [6,7]. This pathway regulates planar cell polarity (PCP) for the morphogenetic processes in vertebrates. ROR1 strongly associates with ROR2, making up the receptor for Wnt5a signaling and activates RhoA through Daam1 in esophageal squamous cell carcinoma (ESCC) and glioblastoma [57,58]. Noncanonical Wnt5a, but not Wnt3a, induces phosphorylation of ROR2 at Ser864 by GSK-3β, leading to cell migration [59]. ROR1 expressed in chronic lymphocytic leukemia (CLL) can also serve as a receptor for Wnt5a, which induces ROR1 to associate with DOCK2 (dedicator of cytokinesis 2) and induces activation of Rac1/2 [60]. Another form of β-catenin-independent noncanonical pathway is the WNT-Ca^2+^-dependent signaling cascade. In this pathway, Wnt stimulates Fzd-mediated activation of heterotrimeric G proteins, which in turn activate phospholipase C (PLC), producing inositol-3-phosphates (IP3). IP3 elicits Ca^2+^ release from intracellular stores in ER and Ca^2+^ activates the effector proteins such as protein kinase C (PKC) that regulate the transcription of genes controlling cell fate and cell migration. This pathway is involved in cancer progression, inflammation and neurodegeneration [61]. Wnt5a, the noncanonical prototype ligand, activates Wnt/Ca^2+^ pathway; by comparison, Wnt3a, the prototypical canonical ligand, activates not only Wnt/β-catenin but also Wnt/Ca^2+^ in the same cell [62]. 

Dishevelled, the cytoplasmic protein directly downstream of Frizzled receptor, interacts with Daam1, a formin domain-containing protein, for Rho activation. Dishevelled binding to WGEF, a Rho-specific exchange factor; this relieves the effects of an autoinhibitory domain in WGEF, leading to Rho activation [63]. In addition, the Rho GEF Trio was recently reported to interact with Dishevelled and activates Rac1 [64]. Constitutively active mutants of Rac1 and Cdc42 potently activate JNK and GEFs as these Rho GTPases selectively stimulate JNK activity. Conversely, expression of inhibitory molecules for Rho GTpases and dominant negative mutants of Rac1 and Cdc42 inhibit JNK activation [65]. 

Although, a detailed mechanism for JNK activation through Rac1 has not been revealed, in general, scaffold proteins link Rac1 activity to JNK activation through the Wnt signaling. This is shown for a direct interaction between JNK1 and CrkII-p130Cas adapter complex being critical for Rac1-induced JNK activation [66]. Similarly, the Rac1/POSH/MLK2,3/MKK4,MKK7/JNK cascade appear to be a mechanism by which Rac1 activates JNK. POSH proteins contain CRIB motifs, which bind to Rac1. Of note, the JNK interacting protein (JIP) group of scaffold proteins interact with multiple kinases, MAPKKK, MAPKK, and JNK [67]. 

### 4.3. Communication between Canonical and Non-Canonical Pathways of Wnt

Canonical and non-canonical pathways are not completely separated, but communicate with one another. In particular, PI3K activates Rac1, which in turn activates JNK2 that phosphorylates β-catenin at Ser191 and Ser605, facilitating β-catenin nuclear translocation in response to Wnt3A [68]. For example, an inactivating mutation in *Drosophila* RacGAP50C promotes canonical Wnt signaling [69]. In addition, Tiam1, a Rac GEF, interacts with β-catenin and Rac, which is promoted by Wnt3a [70]. Being a canonical Wnt prototype ligand, Wnt3a can signal via the β-catenin-independent RhoA-Rac1 pathway to coordinate an embryological event such as gastrulation [71] and activation of RhoA, Rac1 and Cdc42 operates downstream of Dishevelled, linking MEKK1/MEKK4 to JNK activation [72]. Wnt3A also induces not only β-catenin accumulation but also RhoA activation. Tyr42 phosphorylation of RhoA is critical for β-catenin accumulation and nuclear translocation (Figure 3) [46]. 

### 4.4. Regulatory Proteins for β-Catenin Activity

Many protein factors bind to β-catenin (Table 2) and regulate β-catenin transcriptional activity [73]. In absence of Wnt, TCF proteins can bind to their DNA recognition sequence (Wnt response elements: WREs) and recruit Groucho (TLE1 in mammals), a long-range chromatin repressor that functions with histone deacetylases (HDACs), leading to the condensation of local chromatin and inhibition of transcription [74,75]. However, in presence of Wnt, β-catenin accumulates and translocates to nucleus and binds to TCFs instead of Groucho by competing with Groucho; β-catenin contains the central ARM repeats to span R3-R10 of β-catenin contain the TCF binding region [76].

The R10 ARM repeat-C-terminal region of β-catenin also interacts with histone acetyltransferase (HAT), CREB (cAMP-response element binding protein) binding protein (CBP, also known as CREBBP) and p300, resulting in histone acetylation and activation of several Wnt target genes [77,78]. In this detail, p300 acetylates β-catenin at its Lys345 residue in ARM 6 domain, resulting an increase of β-catenin affinity to TCF4, and β-catenin/TCF4 transcriptional activation [79]. KLF4 binds to the C-terminal transactivation domain of β-catenin and inhibits p300/CBP recruitment by β-catenin, inhibiting β-catenin acetylation [80]. Of note, β-catenin is also acetylated at Lys49 [81] by CBP and also the trimethylated Ezh2, and in particular, β-catenin Me3 version acts as a co-repressor of neuronal differentiation genes, Soxq and Sox3 [80]. β-catenin Lys354 is also critical for nuclear retention of β-catenin through a balance of p300 and sirtuins in a glucose dependent manner [82]. The nucleolar protein, block of proliferation 1 (BOP1), also activates Wnt/β-catenin signaling by increasing the recruitment of CBP to β-catenin, enhancing CBP-mediated acetylation of β-catenin [83]. O-GlcNAcylation at Thr41 of β-catenin regulates not only its stability, but also affects its localization in the adherens junction by association with α-catenin [84]. The post-translational modifications of β-catenin are summarized in Figure 4. 

Wnt signaling also induces the interaction between β-catenin and transducing β-like protein 1 (TBL1) and its highly related family member, TBLR1. Depletion of TBL1/TBLR1 significantly inhibits Wnt/β-catenin-induced gene expression and oncogenic growth in vitro and in vivo. The recruitment of TBL1/TBLR1 and β-catenin to Wnt target-gene promoters is mutually dependent. The TBL1/TBLR1 and β-catenin complex replaces TLE1/Groucho and HDAC1, and subsequently binds to TCF [85]. 

Another interaction involves Pygopus protein binding to β-catenin through the linkage of BCL9 [86]. Pygopus also interacts with the mediator complex subunits Med12 and Med13, which recruit general transcription factors to the chromatin [87]. Pygopus interacts with both CBP and H3K4me3, leading to chromatin remodeling [88]. 

Sox (Sry-related HMG box) is a family of key transcriptional factors in animal development [89], and in *Xenopus*, Sox17 and Sox3 bind to β-catenin, competing with TCF/LEF for binding to β-catenin, which results in inhibition of Wnt-signaling [90]. In cultured cells, Sox17 also promotes degradation of TCF and β-catenin and blocks their signaling [91]. Recently, Sox17 and β-catenin were reported to co-occupy hundreds of key enhancers, leading to transcriptional regulation of specific genes. In some cases, Sox17 and β-catenin synergistically activate transcription, independent of TCFs, while for other enhancers, Sox17 represses β-catenin/TCF-mediated transcription [92]. However, for Sox4, it was shown to positively enhance TCF/β-catenin activity [91]. β-catenin also suppresses the expression of Sox9, inhibiting osteoblast differentiation to chondrocytes in the developing skeletal system [93].

For β-catenin-TCF4, hypoxia inhibits their complex formation and transcriptional activity. Under hypoxia, β-catenin directly binds to hypoxia induced factor (HIF)-1α, which is induced and stabilized by low oxygen and acts as a transcription factor for hypoxia-responsive genes. Furthermore, under hypoxia, β-catenin/HIF-1α interaction occurs at the promoter region of HIF-1 target genes, and enhances HIF-1-mediated transcription, thereby promoting cell survival and adaptation to hypoxia [94]. HIF-1α also binds to human ARD1 (hARD1) in hypoxia; as HIF-1α dissociates hARD1 from β-catenin, hARD1 cannot bind to and acetylate β-catenin, preventing β-catenin acetylation and inhibiting TCF4 activity, along with c-Myc suppression and p21 induction. In these ways, HIF-1α inactivates Wnt signaling [95].

Transcription factor Runx2 is crucial for normal bone formation; β-catenin is also essential for proliferation of osteoblasts, a bone forming cell, and normal skeletal development. However, a constitutively active β-catenin enhances LEF1-dependent repression of Runx2 through a mechanism by which LEF-1 interacts with Runx2 and LEF1 represses Runx2-induced activation of mouse osteocalcin 2 promoter [96]. In another scenario, cyclooxygenase-2 (COX-2) overexpression along with downregulation of tumor suppressor 15-prostaglandin dehydrogenase (15-PGDH) and prostaglandin transporter (PGT) in colorectal cancer increases levels of its pro-tumorigenic product prostaglandin E2 (PGE_2_). Here, β-catenin/TCF4 binds to the 15-PGDH and PGT promoters and represses their transcription. Accordingly, β-catenin induces an increase of pro-tumorigenic product PGE_2_ [97]. 

### 4.5. Regulation of β-Catenin Nuclear Translocation

Nuclear translocation of β-catenin may be facilitated by association with APC, BCL9, RAPGEF5, Kinsin2/IFT-A [73]. Interaction of BCL9 with β-catenin requires Tyr142 phosphorylation of β-catenin by Fyn, Fer and C-Met [98]. Another post-translational modification, O-GlcNAcylation of β-catenin, also stimulates its nuclear export [99]. 

With respect to small GTPases, Rac1 and Tiam (a GEF of Rac1) are components of transcriptionally active β-catenin/TCF complex at Wnt-responsive promoters, and the complex of Rac1/Tiam1 serves to enhance target gene transcription [70]. In addition, Rac1 activation induces nuclear translocation of β-catenin, with β-catenin Ser191 and Ser605 which are phosphorylated by JNK [68]. In addition, pyruvate kinase M2 (PKM2) is highly expressed in cancer and contains the nuclear signaling (NLS) residues and it binds to β-catenin, inducing transactivation of β-catenin in response to epithelial growth factor (EGF) [100]. Interestingly, the forkhead box (FOX) transcription factor, FOXM1, also containing an NLS, Ref. [101] binds to β-catenin, promoting β-catenin nuclear import in mammalian cells [102]. Intriguingly, oxidative stresses such as hydrogen peroxide induce the functional interaction between β-catenin and FOXO transcription factors [103]. In addition, Wnt-induced deubiquitylation of FOXM1 ensures nuclear β-catenin transactivation [104]. Other FOX transcription factors regarding to Wnt signaling in cancers have also been described in a previous review paper [105]. Thus, it is likely that the nuclear translocation/activity of β-catenin is regulated by a variety of mechanisms in the context of specific signaling pathways. Furthermore, Parafibromin (in human) and Hyrax (in *Drosophila*), as components of the polymerase associated factor 1 (PAF1) complex, are required for nuclear transductions seen with the Wnt signal and directly bind to ARM repeats (R11-C-terminal region) of β-catenin along with recruitment of Pygopus and BCL9/Legless (Leg) [106] with BCL9/Leg functioning as an adaptor between Pygopus and β-catenin/Armadillo. Pygopus is constitutively localized to the nucleus as a nuclear anchor, leading to nuclear localization of Leg and β-catenin [107].

Recently, it was reported that p-Tyr42 RhoA delivers β-catenin to nucleus via p-Tyr42 RhoA/β-catenin complex formation with the complex detected in the nucleus. The p-Tyr42 RhoA/β-catenin interaction may play a role in transcriptional regulation of specific genes [46]. However, the molecular mechanism by which p-Tyr42 RhoA mediated the translocation to the nucleus remains to be elucidated. Of note, Net1, a RhoGEF, has also been reported to exist in the nucleus, where it activates the nuclear RhoA. Moreover, ionizing radiation specifically promotes activation of the nuclear pool of RhoA in a Net1-dependent manner, while the cytoplasmic RhoA activity is not affected [108]. However, it remains to be shown how RhoA moves to the nucleus by ionizing radiation. The proteins binding with β-catenin are revealed in Figure 5. 

**Table 2 biomedicines-10-01112-t002:** Protein factors binding to β-catenin.

Proteins Binding to β-Catenin	Description	References
p-Tyr42 RhoA	Tyrosine 42 phosphorylated form of RhoA (Ras Homolog Family Member A).	[46]
Rac/JNK2	JNK2 phosphorylates β-catenin at Ser191/605 residues.	[68]
Bcl9/Lgs	Interaction between β-catenin and BCL9 is mediated by the phosphorylation of β-catenin at Tyrosine 142 residue.	[73,106]
Groucho	TLE1 in mammals.	[74,75]
CBP (CREB-binding protein)	Prominent histone acetyltransferase (HAT).	[77,78]
TBL1	Transducing β-like protein 1.	[85]
Sox	Sry-related HMG box, a key transcriptional factor of animal development.	[89]
Sox3	Sox3 binds to β-catenin, inhibiting TCF activity by competing.	[90]
Sox4	Sox4 enhance β-catenin/TCF activity.	[91]
Sox17	Sox17 binds to β-catenin and regulates gene expression.	[90,91,92]
HIF-1α (hypoxia induced factor -1α)	β-catenin enhances HIF-1α-mediated transcription.	[94]
hARD1	hARD1 binds to and acetylates β-catenin, leading to β-catenin activation.	[95]
Runx2	A constitutively active β-catenin enhances LEF1 interaction with and inhibition of Runx2 activity.	[96]
PKM2 (pyruvate kinase M2)	PKM2 transactivates β-catenin upon EGF.	[100]
FoxM1 (Forkhead box protein M1)	FoxM1 promotes β-catenin nuclear import.	[102,104,105]
FOXO (Forkhead box protein O)	ROS induce β-catenin and FOXO interaction.	[103,105]
Parafimbromin	Components of the polymerase associated factor 1 (PAF1) complex.	[106]
Pygopus	Involved in signal transduction through the Wnt pathway.	[106,107]
APC (Adenomatous polyposis coli)	APC is a nuclear-cytoplasmic shuttling protein, and can function as a β-catenin chaperone.	[109]
Connexin43	β-catenin binds to connexin43, p-Tyr265/Tyr313 of connexin43 by Src interfere with the interaction.	[110]
HSP27	Hsp27 interacts with β-catenin, reducing β-catenin-GSK-3β complex.	[111]
Kinesin2/IFT-A (intraflagellar Transport A protein)	Kinesin2 promotes nuclear localization of β-catenin during Wnt signaling.	[112]
RAPGEF5 (Rap Guanine Nucleotide Exchange Factor 5)	RAPGEF5 binding with Rap1a/b plays a role in β-catenin nuclear import.	[113]

## 5. Cross-Talk between Src and Wnt Signaling Pathways

In absence of Wnt, Src remains auto-inhibited and in an inactive conformation with Src bound to canonical LRP5/6 or non-canonical ROR2 receptors in a default state. Once Wnt3a or Wnt5a bind to their receptor, Src is activated with the docking of Src to Dishevelled-2 SH3-binding domain disrupting Src autoinhibition and enabling phosphorylation of Src substrates [114]. As such, Src phosphorylates the tyrosine residue of Fzd2 receptor in response to Wnt3 and Wnt5a [115]. For the Wnt signaling pathways, Src has been reported to bind to Dishevelled-2 at two places, one Src SH3-binding domain and also Src C-terminal domain. By binding, Src phosphorylates Dishevelled-2. In particular, phosphorylation of Tyr18, Tyr27 and Tyr275 of Dishevelled-2 appears to contribute to the ability of Src to enhance Wnt3a/β-catenin signaling. Disruption of Dishevelled-2 SH3-binding domain inhibits LEF/TCF-sensitive transcription. 

With the cell receiving the Wnt3a signal as a canonical Wnt, Src associates with LRP5/6 receptor and phosphorylates Fzd2 receptor at Tyr552, allowing Fyn recruitment and activation. Fyn phosphorylates Stat3 at Tyr705 and also β-catenin at Tyr142, releasing β-catenin from α-catenin and cadherin. Activated Stat3 and β-catenin are then able to act as transcription activators. With the cell receiving the Wnt5a as the non-canonical Wnt, Wnt5a binds to Fzd2 and the co-receptor, tyrosine kinase-like orphan receptor 2 (ROR2). Src constitutively interacts with ROR2 and CK1ε, and p120-catenin also directly binds to ROR2. In a similar manner as with Wnt3a, Src activation by Wnt5a facilitates Dishevelled-2 and Fyn recruitment to Fzd2 receptor. Fyn phosphorylates Stat3 similar to that in canonical signaling and Dishevelled-2 activates Rho GTPases [115]. Of note, superoxide produced through NADPH oxidase in response to Wnt3A has also been reported to activate Src [46]. β-catenin also regulates Src activity with β-catenin inducing PDGF production in pancreatic cancer cells, which stimulates cells in an autocrine manner and leads to Src phosphorylation and activation [116]. 

The positive function of Src in Wnt signaling has some controversy. Both Src and Fer tyrosine kinases associate with LRP6 and directly phosphorylate multiple tyrosine residues of LRP6 including Tyr1460, which negatively regulates LRP6 and Wnt signaling upstream of β-catenin; MEF cells lacking Src and Fer also have enhanced Wnt signaling. In addition, Wnt3a treatment enhances phosphorylation of LRP6 and Src reduces cell surface levels of LRP6. LRP6 tyrosine phosphorylation by Src and Fer is thought to function as negative regulation in this pathway in preventing over-activation of Wnt signaling at the level of the Wnt receptor LRP6 [117].

On interaction of Src with β-catenin, Src phosphorylates β-catenin at Tyr333 residue, which interacts with PKM2, thereby leading to Cyclin D1 expression in response to EGF [100]. In addition, Tyr654 of β-catenin was reported to be phosphorylated by Src in response to hepatocyte growth factor (HGF), leading to release of β-catenin from E-cadherin [118]. Src-mediated Tyr654 phosphorylation of β-catenin enables β-catenin nuclear translocation. Thus, Src inhibitors nintedanib and KX2-391 and Src knockdown inhibited the expression of genes downstream of Wnt signaling such as Cyclin D1, Wisp1 and S100a4 [119]. In addition, Src mainly phosphorylates Tyr86 in β-catenin, which is accompanied with a decrease of interaction with E-cadherin [120]. In primary alveolar epithelial cells, β-catenin phosphorylation at Tyr654 ensures p-Tyr654 β-catenin/p-Smad2 complex formation and initiation of epithelial-mesenchymal transition (EMT) along with α3 integrin requirement with transforming growth factor-β1 (TGF-β1) treatment of the cells [121]. 

An increase in Hsp27 expression facilitates malignant progression of multiple cancers [122]. Of note, Hsp27 interacts with β-catenin, thereby leading to reduced association of β-catenin with GSK-3β and stabilization of β-catenin. In addition, Hsp27 interaction requires β-catenin phosphorylated Tyr654 by Src upon EGF treatment of the cells [111]. The scenario is thought to have unphosphorylated Hsp27 binding to β-catenin and enhancing its phosphorylation by Src. On additional regulation of Hsp27, mitogen-activated protein kinase-activated protein kinase-2 (MAPKAPK2 or MK2) phosphorylates and activates p38 kinase, which in turn phosphorylates Hsp27, keeping Hsp27 from binding to β-catenin. Accordingly, reduction of MK2-mediated Hsp27 phosphorylation by reduced p38 kinase activity from suppression by Her2 in breast cancer cells contributes to β-catenin activation by allowing unphosphorylated Hsp27 to bind to β-catenin; this allows phosphorylation of β-catenin by Src, starting a chain of signaling that leads to dissemination phenotypes in early lesion breast cancer cells [123].

β-catenin induces Connexin43 (Cx43) expression in response to Wnt signaling and furthermore, β-catenin directly interacts with the C-terminal (CT) domain of Cx43 at the gap junctions. However, this interaction is inhibited by Src phosphorylation of Cx43 CT Tyr265 and Tyr313 residues [110]. Moreover, Src interacts with Cx43 and phosphorylates Cx43 Tyr247 and Tyr265, which leads to reduction of Src oncogenic activity and decreases in glioma cell proliferation [124]. Cx43 recruits Src together with Src endogenous inhibitors, C-terminal Src kinase (Csk), which phosphorylates Src Tyr527, and PTEN (phosphatase and tensin homolog), which dephosphorylates Src Tyr416 [125]. Src inhibition was demonstrated by a cell-penetrating Cx43 peptide (Tat-Cx43 266-283) reducing β-catenin expression and increasing astrocytic differentiation of neural progenitor cells [126]. 

Yes-associated protein, a transcription factor (YAP)/transcriptional coactivator TAZ play positive and negative roles in Wnt signaling. YAP/TAZ are integral components of the β-catenin destruction complex that serves as a cytoplasmic sink for YAP/TAZ. In Wnt-activated cells, YAP/TAZ are released from the destruction complex and translocate to the nucleus, thereby leading to Wnt/YAP/TAZ-dependent biological effects. In Wnt-not active cells, YAP/TAZ are required for ubiquitin ligase βTrCP recruitment to the destruction complex and β-catenin inactivation [127]. Regarding Src, the tyrosine kinase negatively regulates Wnt/β-catenin signaling in intestinal epithelial cells in crypts likely through YAP activation [128]. YAP1 was reported to inhibit Wnt signaling, which is the major driving force for homeostatic self-renewal and regeneration in mammalian intestine; transgenic expression of YAP reduces Wnt target gene expression and results in the rapid loss of intestinal crypts. Cytoplasmic YAP restricts elevated Wnt signaling by limiting the activity of Dishevelled through interfering with Dishevelled nuclear translocation during regenerative growth [129]. Of note, YAP/TAZ binds to β-catenin and restricts its nuclear translocation, thereby suppressing Wnt-target gene expression [130]. In contrast, LATS2 that phosphorylates and inhibits YAP is inactivated by its phosphorylation directly induced by v-Src and Src, leading to YAP accumulation and its transcriptional activity [131]. Moreover, Src inhibitors, dasatinib and eCF506, block YAP nuclear localization, indicating YAP nuclear localization is governed by Src [132].

The expression of ROR2, the co-receptor of Wnt5a, was reported to be enhanced in osteoarthritis and ROR2 overexpression inhibits chondrogenesis, whereas ROR2 silencing induces chondrogenesis. Thus, ROR2 blockade enhances cartilage formation and chondrogenic differentiation through YAP. Furthermore, ROR2 inhibits Wnt3A/β-catenin signaling. Of note, Wnt5A treatment in ROR2 overexpressed cells enhances nuclear translocation of YAP by a RhoA-mediated manner [133]. 

Regarding bone formation, activation of canonical Wnt signaling is increased for bone genesis and inhibition of the signaling decreases bone mass and strength [134] as the Wnts signal through both LRP5 and LRP6 co-receptors to enhance bone-forming osteoblast numbers and bone mass [135,136,137]. ROR2 is also expressed in the bone-resorbing osteoclasts and during osteoclast differentiation, Wnt5a and PKN3 are markedly increased. Through Daam2, Wnt5a activates RhoA, which in turn binds to and activates PKN3 an effector protein of RhoA. PKN3 binds to Src and Pyk2 in a ROR2 signaling-dependent manner, leading to Src activation and osteoclastic bone-resorption [138,139]. Regarding ROR2, overexpression of ROR2 increases the metastasis ability of B16 murine melanoma cells, and inhibition of Src is critical for the ROR2-mediated cell migration upon Wnt5a treatment. C-terminus of ROR2 was essential for the mutual interaction with SH1 domain of Src [140].

Recently, it was reported that GSK-3β also regulates Src activity. p-Ser9 GSK-3β has been generally accepted to be an inactive form of GSK-3β kinase in insulin signaling pathway. However, the function of p-Ser9 GSK-3β in Wnt signaling pathway remains undiscovered. It is remarkable that p-Ser9 GSK-3β could phosphorylate Ser43, Ser51, and Ser493 of Src in series with Wnt3A signaling. p-Ser51 and p-Ser493 Src display inactivity along with reduced Src p-Tyr416 levels. However, additional phosphorylation of Ser43 of Src, in turn restores Src activity by restoring its p-Tyr416 levels. Therefore, p-Ser9 GSK-3β harboring a protein kinase activity towards Src regulates Src activity depending on spatially and temporally regulated Wnt3A signaling (Figure 6) [46]. Substrate proteins for Src non-receptor protein Tyr kinase are listed in Table 3. 

## 6. Conclusions and Perspectives

Src associates with LRP5/6 receptor, and is activated by Wnt3a. Activated Src phosphorylates and activates Fzd2 receptor, and Dishevelled-2 phosphorylated by Src can in turn stabilize Src upon Wnt stimulation. β-catenin is phosphorylated by Src at Tyr333 and Tyr654, which enables Src to translocate to nucleus. Moreover, p-Tyr654/p-Tyr86 β-catenin due to Src is readily released from E-cadherin in the membranes. As aberrant Src activation and Wnt signaling pathways are essential for tumorigenesis, the inhibitors to downregulate Src and Wnt signaling have been considered for targeted cancer therapy.

Dasatinib is an ATP-competitive inhibitor for protein tyrosine kinases including BCR-Abl, Src, c-Kit, ephrin receptor and other tyrosine kinases. Accordingly, dasatinib has been used for targeted anti-cancer therapy against chronic myelogenous leukemia (CML) and acute lymphoblastic leukemia (ALL) [142]. Notably, for clinically relevant drug combinations for colorectal cancer, mitogen-activated protein kinase kinase (MEK) and Src inhibition (selumetinib and bosutinib, respectively) was effective when tested ex vivo [143]. Likewise, dual therapy with MEK and Src inhibitors (selemetinib and saracatinib, respectively) potentially inhibited tumor-initiating ovarian cancer cells, in which Src and MAPK both were activated [144]. Main approved src kinase inhibitors as therapeutics and drugs in clinical trials were listed in Table 4. 

Wnt was shown to regulate ovarian cancer pathogenesis with extracellular secreted Wnt signaling inhibitors such as the Dickkopfs (DKKs) and secreted Frizzled related proteins (SFRPs) being considered for anti-cancer protein therapy. DKKs block LDL5/6 from complexing with Fzd [145], and SFRPs bind directly with Wnt ligands or competing with Wnt ligands to bind to the Frizzled receptor [146]. Wnt antagonists including small molecule drugs such as PRI-724, a CBP/β-catenin antagonist, are being developed for targeting the downstream components of the Wnt signaling pathway in ovarian tumorigenesis [147,148]. Current drugs in clinical trials for the components in Wnt/β-catenin signaling pathway were listed in Table 5. Therapeutic targeting of Wnt signaling was also well described in other review papers [147,149]. 

However, blockade of Wnt signaling and non-specific Src inhibitors can cause side effects such as certain tissues being affected, for example, by loss of bone strength and mass [147]. Consequently, to avoid or minimize these side effects, the inhibitors of critical points to link Src and Wnt signaling might be more effective targeted anti-cancer drugs. For instance, Tyr333 and Tyr654 phosphorylation of β-catenin by Src may be actionable points for a targeted inhibitor to block tumorigenesis. 

**Table 4 biomedicines-10-01112-t004:** Approved src kinase inhibitors as therapeutics and drugs in clinical trials.

Drug	Descriptions	Cancer Type	Drug Progression	Identifier (NCT Number)	References
Dasatinib (SPRYCEL^®^)	Dual inhibitor of Src kinase and Bcr-Abl tyrosine kinase	Philadelphia chromosome-positive Chronic myeloid leukemia (Ph + CML), Philadelphia chromosome-positive acute lymphoblastic leukemia (Ph + ALL)	Approved US (2006)	NCT00064233	[150]
Saracatinib (AZD0530)	Dual inhibitor of Src kinase and Abl family kinases1	Alzheimer’s disease	Phase 2a completed (2018)	NCT01864655 (Phase 1) NCT02167256 (Phase 2a)	[151,152]
Bosutinib (SKI-606)	Dual inhibitor of Src kinase and Abl tyrosine kinase	Ph+ Chronic Myeloid Leukemia	Approved US (2012)	NCT00261846 (Phase 2)	[153,154]
Saracatinib (SarCaBon) (AZD0530)	Dual inhibitor of Src kinase and Bcr-Abl tyrosine kinase	Cancer-induced bone pain	Phase 2 completed (2018)	NCT02085603	
KX2-391 (Tirbanibulin) (Klisyri^®^)	Dual inhibitor of Src kinase and tubulin polymerization	Actinic Keratosis (a precursor of squamous-cell carcinoma) on Face or Scalp	Approved US (2020)	NCT03285477 NCT03285490	[155]
TPX-0046	Dual inhibitor of RET receptor tyrosine kinase inhibitor and Src kinase	Non-Small Cell Lung Cancer, Medullary Thyroid Cancer, RET Gene Mutation, Metastatic Solid Tumor, Advanced Solid Tumor	Phase 1/2	NCT04161391	[156]
TPX-0022 (Elzovantinib)	Multi target inhibitor of MET, CSF1R and Src	Non-Small Cell Lung Cancer, Gastric Cancer Advanced Solid Tumor Metastatic Solid Tumors MET Gene Alterations	Phase 1/2	NCT03993873	

**Table 5 biomedicines-10-01112-t005:** Drugs of clinical trials for the components in Wnt/β-catenin signaling pathway.

Description of Mode of Action	Drugs	Cancer Type	Drug Progression	Identifier (NCT Number)	References
CBP/β-catenin antagonist	PRI-724	Advanced pancreatic cancer Metastatic pancreatic cancer Pancreatic adenocarcinoma Advanced solid tumors	Phase 1	NCT01764477 NCT01302405 NCT01606579	[157]
	PRI-724 (with Leucovorin Calcium, Oxaliplatin, or Fluorouracil)	Acute myeloid leukemia Chronic myeloid leukemia	Phase 2	NCT02413853	[158]
FZD10 antagonist	OTSA101-DTPA-90Y	Sarcoma Synovial	Phase 1	NCT01469975	[159]
FZD8 decoy receptor (Fusion protein of FZD8 and IgG Fc domain competing with native FZD8 receptor)	OMP-54F28	Solid tumors	Phase 1	NCT01608867	[160]
	OMP-54F28 (with Nab-Paclitaxel and Gemcitabine)	Pancreatic cancer Stage IV pancreatic cancer	Phase 1	NCT02050178	[161]
	OMP-54F28 (with Paclitaxel and Carboplatin)	Ovarian cancer	Phase 1	NCT02092363	[162]
	OMP-54F28 (with Sorafenib)	Hepatocellular cancer Liver cancer	Phase 1	NCT02069145	[163]
Monoclonal antibody against FZD receptors	OMP-18R5	Metastatic breast cancer Solid tumors	Phase 1	NCT01973309 NCT01345201	[164]
	OMP-18R5 (with Docetaxel)	Solid tumors	Phase 1	NCT01957007	[165,166]
	OMP-18R5 (with Nab-Paclitaxel and Gemcitabine)	Pancreatic cancer Stage IV pancreatic cancer	Phase 1	NCT02005315	[167]
Porcupine inhibitors	CGX1321	Colorectal adenocarcinoma Gastric adenocarcinoma Pancreatic adenocarcinoma Bile duct carcinoma Hepatocellular carcinoma Esophageal carcinoma Gastrointestinal cancer	Phase 1	NCT03507998	[168]
	CGX1321 (with Pembrolizumab)	Solid tumors GI cancer	Phase 1	NCT02675946	[169]
	ETC-1922159	Solid tumor	Phase 1	NCT02521844	[170]
	RXC004	Solid tumor	Phase 1	NCT03447470	[171]
	WNT974	Squamous cell cancer Head and Neck	Phase 2	NCT02649530 NCT01351103	[172]
	WNT974 (with LGX818 and Cetuximab)	Metastatic colorectal carcinoma	Phase 1	NCT02278133	[173]
β-catenin-controlled gene expression inhibitor	SM08502	Solid tumors	Phase 1	NCT03355066	[174]
Wnt signaling pathway inhibitor	SM04690	Osteoarthritis	Phase 2	NCT02536833	[175]
	SM04755	Tendinopathy	Phase 1	NCT03229291	[176]

## Figures and Tables

**Figure 1 biomedicines-10-01112-f001:**
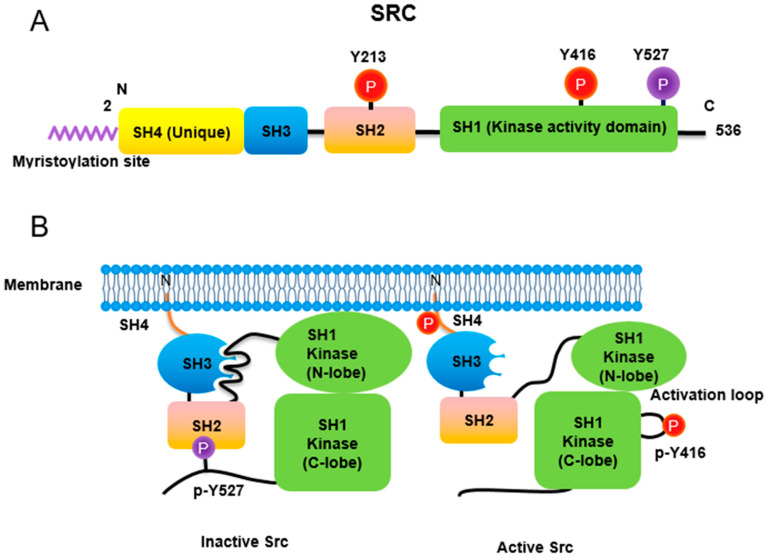
Regulation of Src non-receptor tyrosine kinase activity. (**A**) Src consists of several domains including SH1, SH2, SH3 and SH4 domains. Important Tyr phosphorylation sites of Y527, Y416 and Y213 are noted. (**B**) Tyr527 phosphorylated by C-terminal Src kinase (CSK) binds to SH2 domain in Src, leading to the inhibition of Src activity. Phosphorylated Tyr213 of Src interferes with SH2 domain binding to p-Tyr527 residue of Src. Dephosphorylation of Src Tyr527 by receptor protein tyrosine phosphatase α activates Src and Tyr416 phosphorylation acquires a fully activation.

**Figure 2 biomedicines-10-01112-f002:**
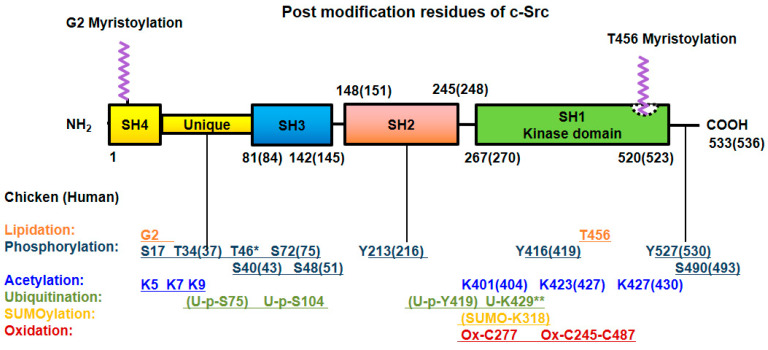
Post-translational modifications of Src. Src undergoes post-translational modification with lipidation, phosphorylation at Ser/Thr and Tyr residues, acetylation, ubiquitylation, SUMOylation and oxidation for its own particular purpose. The numbers of modified amino acid residues in the domains of chicken (human) proteins are denoted. * Referred to chicken only, ** referred to MDCK cell line (*Canis familiaris*, dog) [40].

**Figure 3 biomedicines-10-01112-f003:**
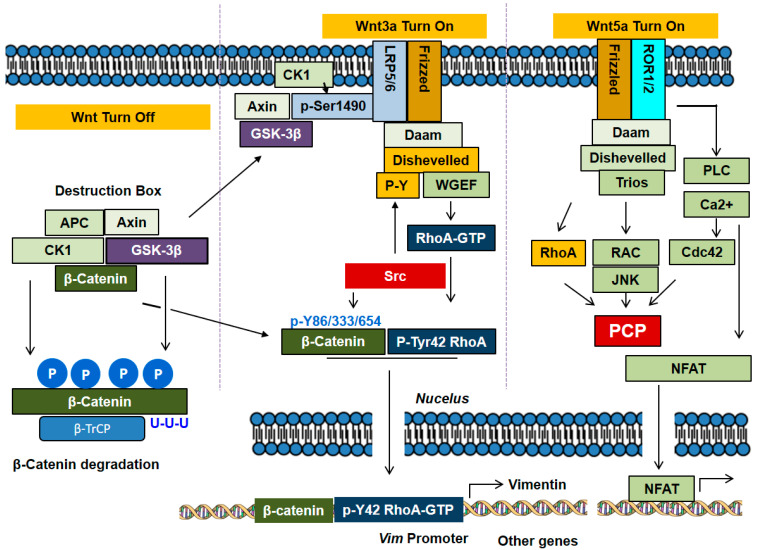
Canonical and non-canonical signaling pathways of Wnts. In absence of Wnt, β-catenin associates with APC, Axin, CK1 and GSK-3β to form the complex of destruction box. CK1 phosphorylates Ser45 in β-catenin, a priming step, and GSK-3β subsequently phosphorylates Ser41, Ser37 and Ser33 in β-catenin, leading to polyubiquitylation by β-TrCP, E3 ubiquitin ligase, and degradation. In the canonical pathway, Wnt3a stimulates the receptors Frizzled and LRP5/6, which recruit Axin and GSK-3β, thereby resulting in dissociation of the destruction box and consequent β-catenin accumulation. In the non-canonical pathway, Wnt5a stimulates the receptors Frizzled and ROR1/2, which induce the activation of Rho subfamily GTPases including RhoA, Cdc42 and Rac1. The Rho GTPases are involved in PCP processes with cell migration. In addition, the non-canonical signaling induces PLC activation and Ca^2+^ mobilization. In particular, in the canonical pathway, Wnt3a activates RhoA and induces Tyr42 phosphorylation of RhoA by Src. Of note, β-catenin forms a complex with p-Tyr42 RhoA, translocates to the nucleus, where the complex regulates the expression of specific genes such as vimentin.

**Figure 4 biomedicines-10-01112-f004:**
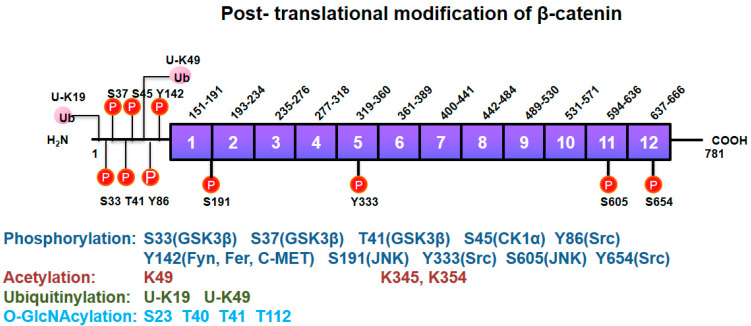
Post-translational modifications of β-catenin. β-catenin undergoes post-translational modifications including Ser/Thr and Tyr phosphorylation, acetylation, ubiquitylation, and O-GlcNAcylation for its own functional regulation. Kinases phosphorylating a specific site in β-catenin are denoted in parentheses.

**Figure 5 biomedicines-10-01112-f005:**
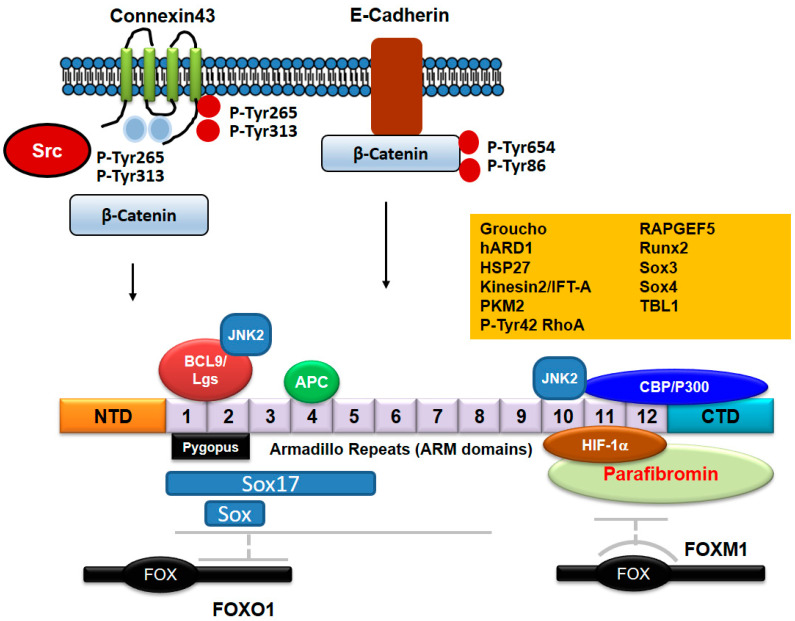
Proteins binding to β-catenin. APC/Axin binds to ARM domains 3-7 of β-catenin. C-terminal domain of FOXO1 binds with ARMs 1-8 of β-catenin, whereas FOXM1 binds to ARMs 11-12 of β-catenin. Pygopus and BCL9/Lgs proteins bind to ARMs 1-2. CBP/P300 binds to C-terminal domain of β-catenin in addition to ARMs 11-12. Sox17 binds to β-catenin ARMs 1-5, but it is not clear that other Sox family proteins bind to which ARMs. JNK2 binds to both ARM domains 2 and 10. HIF-1α binds with β-catenin ARMs 10-12. Parafibromin binds to ARMs 10-12 including C-terminal domain. The exact regions of β-catenin domains binding with other proteins such as Groucho, hARD1, HSP27, Kinesin2/IFT-A, PKM2, p-Tyr42 RhoA, RAPGEF5, Runx2, and TBL1 still remain to be described. β-catenin binds to E-cadherin, but is released from E-cadherin by phosphorylation at Tyr86 and Tyr654 residues by Src. Connexin43 phosphorylated by Src at Tyr654 and Tyr86 residues reduces β-catenin interaction and connexin43 phosphorylated by Src at Tyr247 and Tyr265 residues reduces Src oncogenic activity.

**Figure 6 biomedicines-10-01112-f006:**
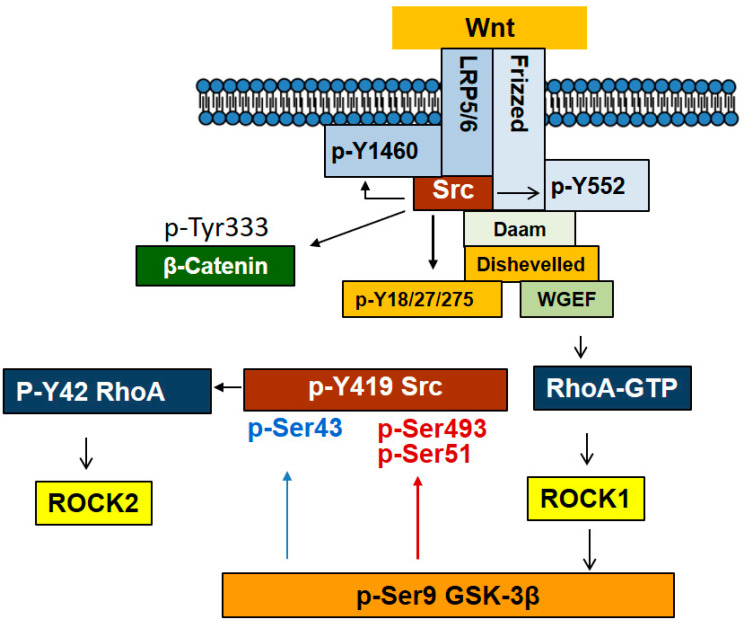
Interaction between Src and Wnt signaling pathway. With Wnt stimulation, Src associates with LRP5/6 and ROR1/2 receptors and is activated. In addition, Src binding to Disheveled ensures Src activation and Src in turn phosphorylates Tyr18, 27, and 275 of Dishevelled, contributing to Wnt signaling. Furthermore, Src phosphorylates Fzd at Tyr552 to stimulate Wnt signaling. However, LRP5/6 Tyr1460 phosphorylation by Src leads to a negative regulation of Wnt signaling. In particular, Src phosphorylates β-catenin Tyr333, facilitating a nuclear translocation along with association with PKM2. Remarkably, p-Ser9 GSK-3β, which is induced by ROCK1, is able to phosphorylate Src at Ser493 and Ser51, leading to Src inactivation while phosphorylating in certain circumstances at Ser43, leading to its activation.

**Table 3 biomedicines-10-01112-t003:** Substrate proteins of Src in Wnt signaling.

Substrate Proteins	Description	References
β-catenin	Src phosphorylates Tyr333 of β-catenin and renders to bind with PKM2.	[100]
	β-catenin Tyr654 phosphorylated by Src is released from E-cadherin.	[118]
	p-Tyr654 β-catenin translocates to nucleus.	[119]
	Src phosphorylates Tyr86 in β-catenin, leading to dissociation from E-cadherin.	[120]
	β-catenin phosphorylation at Tyr654 facilitates binding with p-Smad2 in TGF-β signaling.	[121]
Dishevelled-2	Src phosphorylates Tyr18, Tyr27, Tyr275 and Tyr463 of Dishevelled-2. Src binds to Dishevelled and Dishevelled disrupts Src autoinhibition, thereby Src can phosphorylate its substrate.	[114]
LRP6	Src associates with LRP6 and phosphorylates Tyr1460, leading to feedback inhibition in Wnt signaling.	[117]
Frizzled receptor	Src associated with LRP5/6 receptor phosphorylates Fzd2 receptor at Tyr552.	[115]
ROR2	Interaction of Src with ROR2 is critical for metastasis.	[140]
GSK-3β	Src phosphorylates Tyr216 of GSK-3β, an active form.	[141]

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
