# Peer review of "Cross-Talk between Wnt Signaling and Src Tyrosine Kinase"

_biomedicines, 2022, doi:10.3390/biomedicines10051112_

Round 1

Reviewer 1 Report

In this review authors succinctly summarized the Cross-talk between Wnt signaling and Src tyrosine kinase pathways. However, it would great if authors would be more specific whether this review (Cross-talk between Wnt signaling and Src tyrosine kinase pathway) focused on general biology or specific to cancer or diseases?

At places in the manuscript, font style is different. Please correct it

It might be additional strength for the manuscript, if authors include a table summarizing the what are different inhibitors of Wnt signaling and Src tyrosine kinase currently in clinical trials.

Author Response

  1. Thank you for your constructive comments. We described the general biology on the cross-talk between Wnt-Src first. However, these issues are closely related to the several diseases including cancers. Therein, as you suggested, we summarized the current drugs of clinical trials in Table 4 (Src inhibitors) and Table 5 (Wnt signaling inhibitors).
  2. We complemented the next text in Perspectives. "Notably, for clinically relevant drug combinations for colorectal cancer, mitogen-activated protein kinase kinase (MEK) and Src inhibition (selumetinib and bosutinib, respectively) was effective when tested ex vivo [139]. Likewise, dual therapy with MEK and Src inhibitors (selemetinib and saracatinib, respectively) potentially inhibited tumor-initiating ovarian cancer cells, in which Src and MAPK both were activated [140]. Main approved src kinase inhibitors as therapeutics and drugs in clinical trials were listed in Table 4."

    "Current drugs in clinical trials for the components in Wnt/β-catenin signaling pathway were listed in Table 5. Therapeutic targeting of Wnt signaling was also well described in other review papers [143,145]."

3. As you suggested, we used Arial font in all the text.

Reviewer 2 Report

Extensive biochemistry review of the interactions of Wnt with Src.

This review provides ideas for the development of inhibitors for cancer-targeted therapy, that is, downregulating Src and Wnt signaling pathways, but blocking these signaling pathways will have side effects, so this review emphasizes that critical point inhibitors linking Src and Wnt signaling may be more effective targeted anticancer drugs.

No specific remark: clarity of abstract, text, figures, discussion and conclusion.

Author Response

  1. Thank you so much for your kind and positive comments.